# Engineering Atomic-to-Nano Scale Structural Homogeneity towards High Corrosion Resistance of Amorphous Magnesium-Based Alloys

**DOI:** 10.3390/mi13111992

**Published:** 2022-11-17

**Authors:** Yuan Qin, Wentao Zhang, Kanghua Li, Shu Fu, Yu Lou, Sinan Liu, Jiacheng Ge, Huiqiang Ying, Wei-Di Liu, Xiaobing Zuo, Jun Shen, Shao-Chong Wei, Horst Hahn, Yang Ren, Zhenduo Wu, Xun-Li Wang, He Zhu, Si Lan

**Affiliations:** 1Herbert Gleiter Institute of Nanoscience, School of Materials Science and Engineering, Nanjing University of Science and Technology, Nanjing 210094, China; 2Australian Institute for Bioengineering and Nanotechnology, The University of Queensland, Brisbane 4072, Australia; 3X-ray Sciences Division, Argonne National Laboratory, Lemont, IL 60439, USA; 4College of Mechatronics and Control Engineering, Shenzhen University, Shenzhen 518060, China; 5Suzhou Nuclear Powder Research Institute Co., Ltd., Suzhou 215004, China; 6Institute for Nanotechnology, Karlsruhe Institute of Technology, 76344 Eggenstein-Leopoldshafen, Germany; 7Department of Physics, City University of Hong Kong, Kowloon 999077, Hong Kong SAR, China; 8Center for Neutron Scattering and Applied Physics, City University of Hong Kong Dongguan Research Institute, Dongguan 523000, China; 9Shenzhen Research Institute, City University of Hong Kong, Shenzhen 518057, China

**Keywords:** magnesium-based alloy, liquid–liquid phase transition, corrosion resistance, structural homogeneity

## Abstract

Magnesium-based amorphous alloys have aroused broad interest in being applied in marine use due to their merits of lightweight and high strength. Yet, the poor corrosion resistance to chloride-containing seawater has hindered their practical applications. Herein, we propose a new strategy to improve the chloride corrosion resistance of amorphous Mg_65_Cu_15_Ag_10_Gd_10_ alloys by engineering atomic-to-nano scale structural homogeneity, which is implemented by heating the material to the critical temperature of the liquid–liquid transition. By using various electrochemical, microscopic, and spectroscopic characterization methods, we reveal that the liquid–liquid transition can rearrange the local structural units in the amorphous structure, slightly decreasing the alloy structure’s homogeneity, accelerate the formation of protective passivation film, and, therefore, increase the corrosion resistance. Our study has demonstrated the strong coupling between an amorphous structure and corrosion behavior, which is available for optimizing corrosion-resistant alloys.

## 1. Introduction

Magnesium (Mg) alloys are considered one of the most promising lightweight metals to potentially replace heavier structural materials in the uses of aerospace, marine, and automobile vehicles [1,2]. However, despite the great prospect, Mg alloys still suffer from a number of inherent drawbacks, including the strength–corrosion tradeoff [3,4]. In this regard, many efforts have been devoted to employing amorphous Mg alloys for anti-corrosion uses. Here, an amorphous alloy means that all the metallic atoms in the long range are arranged randomly in the structure. It is found that amorphous magnesium-based alloys are attractive for their high mechanical properties [5,6]. In addition, amorphous alloys also exhibit improved corrosion resistance due to the chemical homogeneity with reduced crystallographic breaks such as grain boundaries, dislocations, and phase segregations, so the chemical attack on those susceptible sites can be largely avoided [7,8]. Nevertheless, for the Mg-based amorphous alloys in particular, the high reactivity of Mg still makes the material vulnerable to corrosion failure [9,10]. To further improve the corrosion resistance of the Mg-based amorphous alloys, previous efforts mainly focus on strategies of surface modifications [11,12] and on alloying corrosion-resistant elements like Nb [13]. Furthermore, forming a passivation film on the alloy surface is also an effective way to prevent corrosion [14,15]. Although various progress has been achieved, the critical barrier has not yet been overcome, which severely hinders the widespread usage of Mg-based amorphous alloys [16,17]. Therefore, more rational and artful ideas for material design down to the nanostructural or atomic level are highly desired for improving the corrosion resistance of the amorphous Mg alloys.

When heating amorphous alloys before melting, typically, several phase transition processes occur in sequence, including separation into amorphous phases with different components [18,19], liquid–liquid phase transition [20,21], and crystallization [22,23]. Among these processes, the liquid–liquid phase transition involves only rearrangement of local structural units and no component variation in the amorphous alloys without a long-range order [24,25]. Although local changes in structure and density are subtle and come with a small amount of heat release, it still makes remarkable differences in mechanical and functional properties [26,27]. Studies show that the liquid–liquid transition is closely related to the thermodynamic stability and mechanical properties of Mg-based amorphous alloys [28]. The thermodynamically-favored local order and loose structure emerge during the liquid–liquid transition, which benefits higher hardness and modulus than a completely disordered structure. In addition, Hu et al. [29] reveal that the metastable state formed during the liquid–liquid transition in (Fe_0.72_B_0.24_Nb_0.04_)_95.5_Y_4.5_ amorphous ribbon could significantly reduce the activation energy and reaction energy barrier of high-temperature oxidization. More importantly, by using synchrotron small-angle scattering, our group has demonstrated that the liquid–liquid phase transition can significantly impact the atomic-to-nano scale homogeneity in the medium-range structure of amorphous Mg alloys [28], which is strongly related to the corrosion resistance of the material [30,31,32,33]. This motivates us to fundamentally understand the correlation between liquid–liquid transition and corrosion, improving the corrosion resistance in amorphous Mg-based alloys.

In this work, by selecting Mg_65_Cu_15_Ag_10_Gd_10_ amorphous alloy as a model material, we reveal that the chloride corrosion resistance of Mg-based alloys could be significantly improved by rearranging the local structural unit in the liquid–liquid transition. By means of potentiodynamic polarization measurement, electrochemical impedance test, immersion test, differential scanning calorimetry, high energy X-ray diffractometry, high-resolution transmission electron microscopy, X-ray photoelectron spectroscopy, and scanning electron microscopy, it is revealed that structural ordering and heterogeneity during the liquid–liquid transition accelerate the formation of the passivation film and therefore improve the corrosion resistance of the Mg-based alloy. This study proves the strong correlation between an amorphous structure and corrosion behavior and provides a new strategy for optimizing corrosion-resistance alloys.

## 2. Experimental Method

### 2.1. Sample Preparation

The target material studied in this paper was Mg_65_Cu_15_Ag_10_Gd_10_ amorphous alloy (by atomic percentage), with the purity of raw materials being 99.95 wt.%. In order to ensure uniform mixing of each component, the total metal mass was controlled for no more than 10 g. The surface oxide of raw materials was removed by file and sandpaper, and the mass error of weighing did not exceed 0.001 g.

The alloy ingot was prepared by induction heating and then vacuum arc melting many times. The chamber vacuum was below 1 × 10^−3^ Pa and then filled with high-purity argon gas at −0.5 Pa as protection. Then, the quartz tube containing the alloy ingot was placed inside a copper induction coil and electrified by a current of 50 A. Due to the pressure difference, the molten alloy ingot was ejected through a 0.9 mm hole onto the rotating copper roller at a high speed of 60 rps. Eventually, the amorphous ribbons were obtained. A few of them were heated to the temperature of liquid–liquid transition (i.e., T_C_) (444 K) in the oven, hereafter referred to as the Tc-treated samples. The Tc is determined by the temperature at an abnormal exothermic peak in differential scanning calorimetry (DSC), which will be discussed later.

### 2.2. Corrosion Resistance Test

Prior to the corrosion experiments, both as-cast and heated ribbon specimens were cut to 10 mm × 2 mm, fixed in epoxy resin, polished with 2000 mesh sandpaper, and then sonicated in acetone, ethanol, and deionized water, respectively. The electrochemical experiments were carried out using the CHI660E A20146 electrochemistry workstation. The reference electrode and counter electrode were saturated calomel electrode (SCE) and Pt, respectively. The potential polarization scanning was started at 150 mV below the open circuit potential (OCP) at a rate of 0.833 V·s^−1^ in both 1 mol·L^−1^ NaCl and 1 mol·L^−1^ NaOH solutions. Electrochemical impedance spectroscopy (EIS) was performed at the frequency ranging from 10^5^ to 10^−2^ Hz with an amplitude of 5 mV. The solutions of the immersion test lasting 2 h were 0.01 mol·L^−1^ NaCl and 0.01 mol·L^−1^ NaOH.

### 2.3. Characterization Techniques

A total of 20 mg Mg_65_Cu_15_Ag_10_Gd_10_ ribbon was used to test the thermophysical parameters by differential scanning calorimetry (DSC, METTLER TOLEDO) at a heating rate of 10 K·min^−1^. The nitrogen flow rate was set to 50 mL·min^−1^ to isolate oxygen and prevent drastic oxidation of the sample during the heating process. High-resolution transmission electron microscopy (HRTEM, FEI TECNAL G2 20), small-angle synchrotron X-ray scattering (SAXS, Advanced Photon Source of Argonne National Laboratory), and high energy X-ray diffractometry (HEXRD) were used to probe the alloy microstructure. The thin region of the HRTEM sample was obtained by etching the amorphous ribbon with an incident angle of 8° and an energy of 8 keV at low temperature of liquid nitrogen for 25 min. The beam size of the synchrotron X-ray (λ = 0.0886 nm) was 0.1 × 0.2 mm and the SAXS data with Q range from 0.0034 to 0.364. Å^−1^ was calibrated and corrected by empty cell scattering, transmission, and detector response using a beamline MATLAB program package [34]. The incident wavelength of the HEXRD excited by Ag K*α*1 was 0.05594 nm (22 keV), and the scattering angle (2*θ*) ranged from 5° to 20°. The surface morphology of the samples was characterized by scanning electron microscopy (SEM, JSM-IT500HR) with an electron beam voltage of 20 kV. The surface composition and valence state of the alloy were analyzed by X-ray photoelectron spectroscopy (XPS, Thermo Fisher Scientific K-Alpha) with monochromatic Al K*α* ray (h*v* = 1486.6 eV) radiation.

## 3. Results and Discussion

### 3.1. Microstructure and Thermal Analysis

The differential scanning calorimetry (DSC) curve of Mg_65_Cu_15_Ag_10_Gd_10_ amorphous alloys (Figure 1a) shows an abnormal exothermic peak that appeared after the glass transition temperature T_g_ (422 K) [35]. The corresponding temperature was marked as T_C_ (444 K), which, according to previous studies, is related to the liquid–liquid phase transition [28,36]. As the temperature continues to increase, a sharp exothermic peak starting at 458 K emerges, which can be assigned to the crystallization of the amorphous material [37]. The width and thickness of the ribbons are 2 mm and 45 μm, respectively (Figure 1a inset).

From the HRTEM image of the T_C_-treated sample (Figure 1b), the atoms are arranged in a maze-like disordered manner, whereas the selected area electron diffraction (SAED) pattern in the inset shows broad and dispersive rings. The results further demonstrate that the Mg_65_Cu_15_Ag_10_Gd_10_ alloys remain the amorphous feature after T_C_ treatment.

Down to the nanoscale, an interference peak occurs in the T_C_-treated sample whereas it does not exist in the as-cast sample according to the results of SAXS (Figure 1c), which suggests the occurrence of nanoscale heterogeneity different from the amorphous matrix in the T_C_-treated sample [28,38,39,40]. Furthermore, a spheroid model with polydispersity is employed to fit the SAXS profile of the T_C_-treated sample, and the obtained size distribution function result is displayed in Figure 1c inset [41,42]. It is seen that the diameter of the spheroidal granular-like structure is around 5.4 nm for the T_C_-treated sample, verifying that the heterogeneous structure exists [43,44,45]. Further down to the atomic scale, Figure 1d compares the structure factor (S(*Q*)) patterns as a function of momentum transfer amplitude (*Q*) for both the as-cast and T_C_-treated alloys. The first S(*Q*) peak shows a lower peak position and a narrower peak width after T_C_ treatment, which indicates that the local order is extended to the medium range and the atomic packing is less dense than the matrix. The structural correlation is enhanced from the short to the medium range (~5–10 Å) during this process, due to the recombination of the local atomic units [22,46,47]. To summarize, the nanoscale thermodynamically-favored metastable amorphous heterogeneous region with an average diameter of 5.4 nm is uniformly distributed in the amorphous alloy matrix. In each heterogeneous region, the degree of the atomic order is extended to the medium range (~5–10 Å). The heterogeneity in the atomic-to-nano scale stands for the liquid–liquid transition and the strong connection between abnormal exothermic peak at T_C_ and liquid–liquid transition is consistent with our previous studies [28,48,49].

### 3.2. Analysis of Potentiodynamic Corrosion Behavior

Figure 2a shows the potentiodynamic polarization curves of the Mg_65_Cu_15_Ag_10_Gd_10_ as-cast and T_C_-treated samples in 1 mol·L^−1^ NaCl solution, showing an active dissolution process through anodic polarization. By using the Tafel epitaxial method, the self-corrosion potential of the as-cast and T_C_-treated samples were −1.14 V and −1.11 V, respectively, and the corresponding self-corrosion current densities were 1.1 × 10^−3^ A·cm^−2^ and 5.3 × 10^−4^ A·cm^−2^, respectively. The higher corrosion potential and lower corrosion current of the T_C_-treated sample indicate that the chemical stability in the NaCl solution is enhanced after the T_C_ heat treatment. Figure 2a inset shows the curve of open circuit potential of the as-cast and T_C_-treated samples as a function of immersion time in 1 mol·L^−1^ NaCl electrolyte. In the first 100 s, the OCP of the amorphous ribbons changes rapidly to positive, indicating that their stability increases due to the construction of the surface film. Subsequently, the OCP curves of these two samples gradually become stable, with a slight fluctuation in a certain potential, which is attributed to the loss and regrowth of the passivation layer on the surface. During the last 300 s immersion, the OCP of the T_C_-treated sample is higher than that of the as-cast sample, indicating that a better defense and stability layer is formed on the surface of the T_C_-treated sample.

Figure 2b shows the potentiodynamic polarization curves of the as-cast and T_C_-treated Mg_65_Cu_15_Ag_10_Gd_10_ samples in 1 mol·L^−1^ NaOH solution, both of which exhibit good chemical stability with a corrosion current of about 1 × 10^−5^ A·cm^−2^ and a corrosion potential of −1.13 V. A spontaneous passivation occurs for both of the materials with a low passivation current density of about 3 × 10^−4^ A·cm^−2^. Although both the materials show a wide passivation zone (−1.10~−0.98 V for the as-cast sample vs. −1.10~−0.90 V for the T_C_-treated sample), the passivation film breakdown potential of the T_C_-treated sample is slightly higher, indicating that the passivation film of the T_C_-treated sample is more stable [50]. When applying the potential to above −0.90 V, the passivation films of these two samples are transpassively dissolved, and accordingly, the corrosion current density increases [51]. All the above results indicate that the Mg_65_Cu_15_Ag_10_Gd_10_ amorphous alloy system shows better corrosion resistance after T_C_ treatment in both NaCl and NaOH solutions.

Figure 2c,d show the EIS fitting circuits in 1 mol·L^−1^ NaCl and 1 mol·L^−1^ NaOH solutions, respectively. The fitted circuit elements include solution resistance (R1), charge transfer resistance (R2), and constant phase element (Q), and the fitted results are shown in Table 1. After T_C_ treatment, the corrosion behavior does not change compared with the as-cast one, and all the EIS curves in NaCl and NaOH solutions show a single capacitive arc with one time constant [52]. From the fitting results, the R2 values of the T_C_-treated samples are higher than those of the as-cast samples in both NaCl and NaOH solutions. The larger R2 is inclined towards a higher corrosion potential, lower corrosion current, and higher passivation film breakdown potential, which is consistent with the potentiodynamic polarization results (Figure 2a,b) [53].

### 3.3. Analysis of Immersion Corrosion Behavior

Figure 3 shows the SEM images of the as-cast and T_C_-treated Mg_65_Cu_15_Ag_10_Gd_10_ metal glass ribbons soaked in 0.01 mol·L^−1^ NaCl and 0.01 mol·L^−1^ NaOH solutions for 2 h. The passivation films of the as-cast ribbon in 0.01 mol·L^−1^ NaCl show local corrosion and cracks (arrows in Figure 3a), which are largely inhibited in the T_C_-treated sample (Figure 3b). The higher corrosion resistance of the T_C_-treated sample also verifies the experimental results of potentiodynamic polarization curves and EIS. Analogously, the corrosion resistance of the T_C_-treated Mg_65_Cu_15_Ag_10_Gd_10_ ribbon in 0.01 mol·L^−1^ NaOH solution is a little better than that of the as-cast sample, which is inferred by the disappearing pits after the T_C_ treatment.

In order to better understand the chemical stability of the Mg_65_Cu_15_Ag_10_Gd_10_ amorphous alloys, XPS measurements were performed on the as-cast and T_C_-treated ribbons immersed in 0.01 M NaCl and 0.01 M NaOH for 2 h (Figure 4a,b). The relevant elemental content obtained from XPS is displayed in Table 2. For the as-cast samples, the Ag content is lower than the designed composition, which is related to the highest antioxidative activities of Ag [54]. Remarkably, the Mg content is much higher than the designed ratio, indicating the passivation film on the surface of the ribbon is dominated by MgO. When subjected to the T_C_ treatment, the Mg content on the surface of the T_C_-treated sample relative to the as-cast sample decreases remarkably in NaCl. This phenomenon firmly validates the improved corrosion to NaCl for the T_C_-treated alloy. For the samples in NaOH, the Mg content of the T_C_-treated sample is similar to that of the as-cast sample. This is because of the higher corrosion resistance of the prepared alloy in NaOH. Chloride ions are highly corrosive because they, with a small radius and strong permeability, could penetrate the relatively loose passivation film into the matrix and act as ionic conductors to accelerate the anodic polarization corrosion of alloy [55]. Figure 4c,d show the corresponding high-resolution Mg 1 s spectra. The peak located at 1303.57 eV corresponds to Mg metal [56], whereas the peak at 1304.36 eV can be assigned to Mg^2+^ [57]. A lower Mg^2+^ content is detected in the T_C_-treated sample by comparing the XPS spectra of NaCl-soaked ribbons, indicating that the T_C_ heat treatment could effectively protect Mg from oxidation corrosion.

### 3.4. Discussion

Replacing Cu in the traditional ternary Mg_65_Cu_25_Gd_10_ amorphous alloy with Ag element to form a quaternary Mg_65_Cu_15_Ag_10_Gd_10_ one was an excellent way to improve corrosion resistance because of the higher equilibrium electrode potential of Ag, namely better chemical stability, than that of Cu [58]. Our experimental results of electrochemical and microscopic tests, as well as XPS, confirm the excellent alkali corrosion resistance of the amorphous Mg_65_Cu_15_Ag_10_Gd_10_ alloy system.

The improvement in chemical stability, i.e., corrosion performance, could be related to the thermodynamically-favored change in microstructure brought by the liquid–liquid transition. During the liquid–liquid phase transition, a medium-range ordered structure shows higher thermodynamic stability than the totally disordered structure. So, spontaneously, combinations of local atomic units occur, forming nanoscale “medium-range-ordered island” distributed heterogeneously within the amorphous matrix [59,60,61]. Driven by this structural change, the electrochemical reaction of micro galvanic cells formed by the fluctuation of surface energy is small in area and large in quantity, so the formation of passivation film is accelerated. In this micro galvanic cell, the alloy matrix acts as a cathode, whereas the amorphous heterogeneous structure acts as an anode. The area of the cathode matrix is much larger than that of the active amorphous heterogeneous structure anode, which ensures the electrochemical reaction of the passivation film generation at the early stage of corrosion. To sum up the above, the corrosion is accelerated at the beginning of the corrosion process in the T_C_-treated sample. As a result, at the beginning of the corrosion, the T_C_-treated sample could evenly form more corrosion products quickly. When the formation rate of the passivation film is higher than the dissolution rate, compact and uniform passivation film would be formed to prevent the solution from entering the matrix and causing further corrosion, meaning that the alloy possesses good chemical stability. The higher R2 value of the EIS results of the T_C_-treated sample clearly demonstrates that the passivation film of the T_C_-treated sample is not prone to be dissolved. Note that the liquid–liquid transition does not result in structural incoherence, such as grain boundary and dislocation, so the T_C_-treated sample is less likely to corrode by a network [62].

## 4. Conclusions

This paper reports a new approach to improve the corrosion resistance of a typical Mg-based amorphous alloy Mg_65_Cu_15_Ag_10_Gd_10_ of an anomalous exothermic peak by heating the material to the T_C_ temperature of liquid–liquid transition. During this transition, the structural homogeneity slightly decreases with the rearrangement of the local structural units, which accelerates the formation of protective passivation film and therefore increases the corrosion resistance of the Mg-based alloy. Our present study has initiated a new route to optimize corrosion resistance by engineering atomic-to-nano scale structural homogeneity, which could be applied to other amorphous alloy systems with the liquid–liquid phase transition and hopefully extended to widespread corrosion-resistant alloy research.

## Figures and Tables

**Figure 1 micromachines-13-01992-f001:**
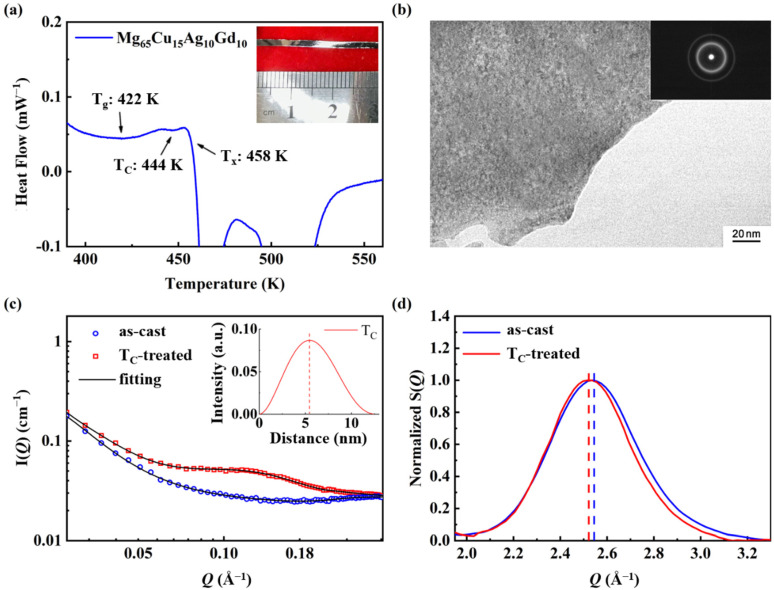
(**a**) Differential scanning calorimetry curve of Mg_65_Cu_15_Ag_10_Gd_10_ amorphous alloy ribbon. The inset is an optical image of an as-cast sample. (**b**) High-resolution transmission electron microscopy of Mg_65_Cu_15_Ag_10_Gd_10_ T_C_-treated ribbon. The inset is the selected area electron diffraction pattern. (**c**) The SAXS of Mg_65_Cu_15_Ag_10_Gd_10_ as-cast and T_C_-treated ribbons. Reproduced with permission [28]. Copyright 2021, Springer Nature. The inset is the size distribution function based on the SAXS profiles. (**d**) The structure factor of Mg_65_Cu_15_Ag_10_Gd_10_ as-cast and T_C_-treated ribbons.

**Figure 2 micromachines-13-01992-f002:**
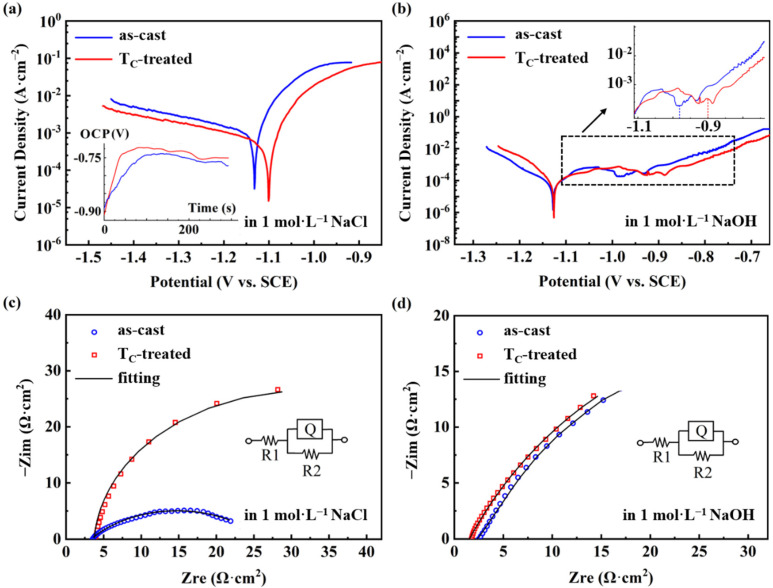
(**a**) Potentiodynamic polarization curves of as-cast and T_C_-treated samples in 1 M NaCl solution. The inset is their open circuit potential (V vs. SCE) curves dependent on the immersion time. (**b**) Potentiodynamic polarization curves of as-cast and T_C_-treated samples in 1 M NaOH solution. The inset enlarges the breakdown potential with the same coordinate units. (**c**) Nyquist curves of as-cast and T_C_-treated samples in 1 M NaCl solution. (**d**) Nyquist curves of as-cast and T_C_-treated samples in 1 M NaOH solution.

**Figure 3 micromachines-13-01992-f003:**
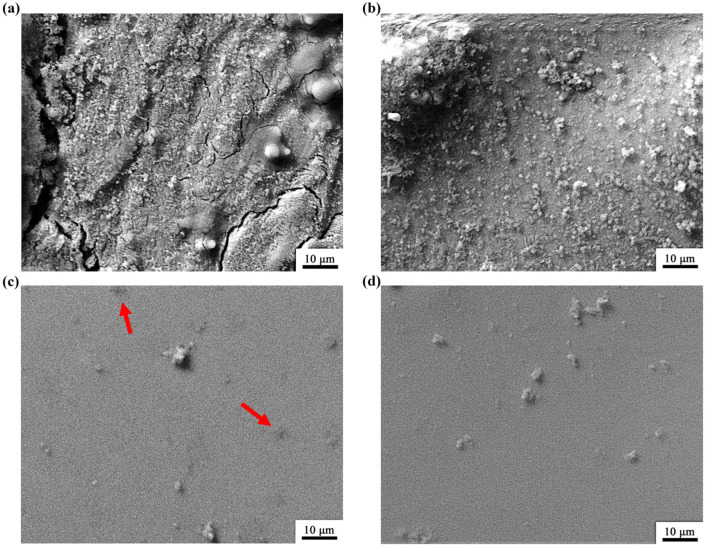
SEM images of (**a**) as-cast and (**b**) T_C_-treated samples after a 2-h immersion in 0.01 M NaCl solution. SEM images of (**c**) as-cast and (**d**) T_C_-treated samples after a 2-h immersion in 0.01 M NaOH solution. The red arrows in Figure 3c show the corrosion pits induced by the NaOH solution.

**Figure 4 micromachines-13-01992-f004:**
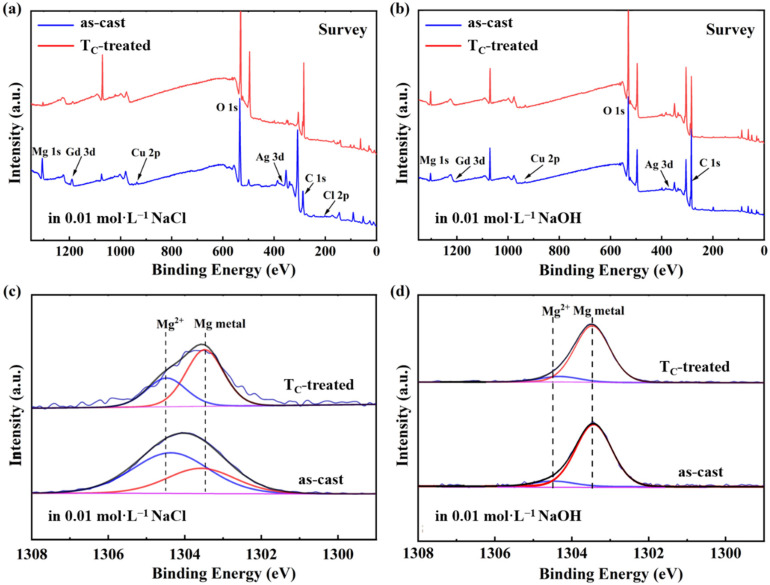
(**a**) Surface XPS of as-cast and T_C_-treated samples after a 2-h immersion in 0.01 M NaCl solution. (**b**) Surface XPS of as-cast and T_C_-treated samples after a 2-h immersion in 0.01 M NaOH solution. (**c**) Mg 1 s spectra of as-cast and T_C_-treated samples after a 2-h immersion in 0.01 M NaCl solution. (**d**) Mg 1 s spectra of as-cast and T_C_-treated samples after a 2-h immersion in 0.01 M NaOH solution.

**Table 1 micromachines-13-01992-t001:** EIS fitting parameters of as-cast and T_C_-treated samples of all the same size in 1 M NaCl and NaOH solutions.

	Q (μs^0^/Ω)	α	R1 (Ω·cm^2^)	R2 (Ω·cm^2^)
as-cast in 1 M NaCl	421.1	0.56	2.77	24.00
T_C_-treated in 1 M NaCl	1000.0	1.00	3.70	56.06
as-cast in 1 M NaOH	545.1	0.65	2.51	81.72
T_C_-treated in 1 M NaOH	77.99	0.77	2.45	84.24

**Table 2 micromachines-13-01992-t002:** Atomic ratios of elements on the surfaces of as-cast and T_C_-treated samples after a 2-h immersion in 0.01 M NaCl and NaOH solutions.

	Mg	Cu	Ag	Gd	O	Cl	Total
as-cast in 0.01 M NaCl	13.49	1.32	0.25	0.31	83.02	1.61	100
T_C_-treated in 0.01 M NaCl	3.49	0.66	0.33	0.18	93.77	1.57	100
as-cast in 0.01 M NaOH	8.04	1.54	0.56	0.31	89.55	-	100
T_C_-treated in 0.01 M NaOH	11.07	1.14	0.43	0.16	87.20	-	100

## Data Availability

The data that support the findings of this study are available from the authors upon reasonable request.

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
