# Peer review of "Engineering Atomic-to-Nano Scale Structural Homogeneity towards High Corrosion Resistance of Amorphous Magnesium-Based Alloys"

_micromachines, 2022, doi:10.3390/mi13111992_

Round 1

Reviewer 1 Report

In this work, the authors have made a good attempt to improve the corrosion resistance by heating the material to the TC temperature of liquid-liquid transition. An amorphous alloy Mg65Cu15Ag10Gd10 with anomalous exothermic peak has been selected a model material. Various advanced characterization techniques such as DSC, synchrotron high-energy XRD, SEM, high-resolution TEM, and XPS were employed to demonstrate that the atomic to nanoscale homogeneity affected by the liquid-liquid phase transition plays an important role in improving the corrosion resistance of Mg-based alloys. This work not only provides a new strategy for optimizing corrosion-resistant alloy processes, but also demonstrates the strong coupling between amorphous structure and corrosion behavior, which is favorable for optimizing corrosion-resistance alloys. So, the reviewer considers that this manuscript is well-prepared for the publication in Micromachines. For the sake of a perfect work, a minor revision is required before publication.

(1) The temperature labeled in Fig. 1a is not consistent with the temperature described in the main text. Please check it.

(2) In Fig. 2b, it is suggested to enlarge the difference of breakdown potential between the two curves.

(3) Fig. 3c and Fig. 3d seems very similar to me. Please explain the difference in the main text.

(4) Do the authors think the method proposed in this manuscript can be extended to other amorphous alloys for improving the corrosion property?

Reviewer 2 Report

The paper proposed a new strategy to improve the corrosion resistance of amorphous Mg-based alloys. The authors reported that the liquid-liquid transition of Mg65Cu15Ag10Gd10 alloys could rearrange the local structural unites in the amorphous structure, slightly decreasing the alloy structures homogeneity, accelerates the formation of protective passivation film and thus increases the corrosion resistance. The authors demonstrated the strong coupling between amorphous structure and corrosion behavior, which can be beneficial to optimize the corrosion-resistance alloys. This work is recommended to publish in Micromachines after the following points being addressed carefully.

1.      In the 2nd paragraph of Introduction, it was mentioned “…separation of the amorphous phase…”. Here the authors are suggested to clarify the amorphous phase according to definition of phase, and precise statements for the amorphous phase should be provided.

2.      In Section 2.1 of Experimental Method, it mentioned that “A few of them were heated to the temperature of liquid-liquid transition (Tc) (443 K)”. The authors are suggested to make further explanation of the corresponding transition temperature.

3.      The authors are suggested to keep consistent on the terminology of “amorphous alloys” and “metallic glass” throughout the entire manuscript.

4.      The authors mentioned the “thermal stability” and “chemical stability” in the manuscript. Detailed discussion and analysis on these different stabilities with respect to the structure and properties of amorphous Mg65Cu15Ag10Gd10 alloys should be provided.

5.      In Section 3.4 of Discussion, it was mentioned that “The improvement of corrosion performance could be related to the change of microstructure brought by liquid-liquid transition, forming nanoscale metastable amorphous precipitates distributed uniformly in the amorphous alloy matrix.” Further evidences are required to confirm the formation of nanoscale metastable amorphous precipitates distributed uniformly in the amorphous alloy matrix.

Reviewer 3 Report

In this manuscript the authors studied the influence of a liquid-liquid transition on the corrosion resistance of Mg65Cu15Ag10Gd10 amorphous alloy. This manuscript has been well organized, and the discussion is clear. This study has demonstrated the relationship between amorphous structure and corrosion behavior in amorphous alloys and provides a new approach for optimizing the corrosion resistance of amorphous alloys. Therefore, this work is of interest and significance, and thus I recommend publishing this manuscript after the following revisions.

1) Line 132-134, the data given here are different from those marked in Fig. 1(a). Please keep them consistent.

2) Line 137-140, it needs to be clear whether the TEM observation here is for as-cast or heated samples. Additionally, the authors refer to “Tc-treated” in the text and the caption of Fig.1 (Line 160). For clarity, the authors need to explain it before.

3) Line 169, I suggest that the data of the self-corrosion current density should also be given here.

4) In the section of 3.4 Discussion (Line 252-279), the discussion here is basically reasonable. However, lack of direct characterization of heterogeneous structure caused by Tc treatment makes the relevant discussions seem inadequate. Therefore, if possible, it is better to provide the information on the structure of the samples.

5) There are a few grammar errors as follows.

(1)    The definite article of “the” is missed in many places. for example, in Line 198, “of TC-treated samplesshould be changed to “of the TC-treated samples”; in Line 231, “subjected to TC treatment” should be changed to “subjected to the TC treatment”.

(2)    Line 199, “higher than the as-cast samples” should be changed to “higher than that of the as-cast samples”.

(3)    Line 233-234, this sent”nce is not clear, and is suggested to modify.

(4)    Line 256, “chemical stability, than Cu” should be changed to “chemical stability, than that of Cu”.

Round 2

Reviewer 2 Report

The authors addressed most ssues which are highlighted by the referees.
I think this paper is ready to publish after the below issue has been solved.
Relevant discussion and analysis on the thermal stability and chemical stability can be provided.
